# The Effectiveness of a CrossFit Training Program for Improving Physical Fitness of Young Judokas: A Pilot Study

**DOI:** 10.3390/jfmk7040083

**Published:** 2022-10-08

**Authors:** Arman V. Avetisyan, Ashot A. Chatinyan, Aspen E. Streetman, Katie M. Heinrich

**Affiliations:** 1Armenian State Institute of Physical Culture and Sport, Yerevan 0070, Armenia; 2Department of Kinesiology, Kansas State University, Manhattan, KS 66506, USA

**Keywords:** motor skills, testing, experiment, social survey, judo, youth sport

## Abstract

The aim of this pilot study was to examine the effectiveness of a CrossFit-based training program to enhance the general and sport-specific physical fitness of 10–12-year-old judokas. The study was conducted between September 2021 and February 2022. The pedagogical research experiment was designed to be one complete, annual macrocycle (September–June). The current study presents mid-point data. Twenty male participants (3 years average sports experience; age = 11 ± 0.64 years) were randomly assigned to one of two groups: CrossFit-based training (experimental, *n* = 10) and traditional training (control, *n* = 10). Baseline testing was conducted by the researcher and included tests for motor skills and general physical fitness domains including Sweden wall pull-ups and leg raises, push-ups, long jump, squats, burpees, shuttle run, and forward rolls. Judo-specific tests included O Soto Gari and O Goshi throws. CrossFit-based training was implemented twice per week for 15–20 min in the experimental group after usual training. The control group completed traditional methods of physical fitness training for judokas with the same training load regarding time. Experimental group participants significantly improved on leg raises (*p* < 0.01), push-ups (*p* < 0.05), and shuttle run (*p* < 0.001); the control group improved their shuttle run (*p* < 0.001). Only the experimental group improved on the O Soto Gari (*p* < 0.01) and O Goshi throws (*p* < 0.05). Results showed that the use of CrossFit-based trainings had a positive effect on 10–12-year-old judokas’ speed-strength abilities, speed-strength endurance, and muscular strength.

## 1. Introduction

In judo, as in all sports, athletes require a high level of physical fitness to be competitive. Given this need, much effort is focused on improving training protocols and the structure of individual training [1]. Coaches are compelled to look for innovative strategies that enhance training and recovery. Such strategies are especially critical in a landscape where competition rules often change each year, requiring the need for adapting training strategies to increase athlete competitiveness. As well, athletes are highly competitive at increasingly young ages. Some experts believe that traditional training systems are outdated and emphasize the requirement for obvious modernization using current methods and means [2]. For example, sports games, strength exercises with weights, gymnastics and acrobatic exercises, long-distance running, and jumping exercises are most often used for conditioning [2]. CrossFit is a relatively new training modality that many experts advocate to train young judokas [2,3,4,5,6]. According to its founder, Greg Glassman, “CrossFit is constantly varied, high-intensity functional movement” that incorporates a variety of exercises together to train multiple energy systems and fitness domains [7]. 

Previous research demonstrates the physical benefits of CrossFit training for students [8,9,10,11,12,13], police officers [14], military personnel [15,16], and athletes in various sports [3,7,17]. More limited research indicates CrossFit may be a promising training protocol among younger students. A randomized control trial of middle school aged students comparing CrossFit Kids (CFK) to traditional PE classes demonstrated physical and mental health improvements [18]. While students in both conditions increased in physical fitness over the course of the study, those in the CFK condition reported more strength and confidence gains [18]. In contrast, Eather et al. [19] determined that participation in the CrossFit Teens did not improve mental health outcomes. While the study did provide preliminary evidence for improving mental health in adolescents ‘at risk’ of developing psychological disorders, more research is needed to better understand the effects of CrossFit participation among youth.

One study demonstrated the effectiveness of regular CrossFit workouts on judokas aged 16–17 years [6]. In the research, emerging evidence suggests that the use of regular CrossFit-based trainings in the practice of young judokas significantly increased their combat activity in competitive matches. Athletes who used CrossFit training during the pre-competition and competitive training periods won a higher percentage of matches than athletes of the control group (with an average difference of more than 5%). CrossFit research on younger athletes aged 10–12 years is limited despite evidence that this is the best age to develop many motor skills [20,21,22]. Further research is required to understand the effectiveness of using CrossFit in this age category. 

When assessing fitness among younger judo athletes, the use of tests reflecting multiple fitness components is recommended [23,24]. This more-comprehensive fitness approach allows for identifying weaknesses and for developing ways to improve them [23]. There are also recommended judo-specific tests, including the O Soto Gari throws, as suggested by Khomichev and Tarakanov [2].

This pilot study research aimed to examine the effectiveness of a CrossFit-based training program to enhance the general and sport-specific physical fitness of 10–12-year-old judokas. We hypothesized that young judokas randomly assigned to the CrossFit training condition would demonstrate greater improvements in general and sport-specific physical fitness compared to the control condition.

## 2. Materials and Methods

### 2.1. Experimental Approach to the Problem

The pedagogical research experiment was held at the “Youth Sports and Cultural Training Center after Vahe Zakaryan” in Hrazdan, in the Republic of Armenia. The pilot study was conducted between September 2021 and February 2022. The intervention was designed to be one complete, annual macrocycle (September–June). However, the current study presents the mid-point data. By examining mid-point data, it was possible to identify the initial impact of CrossFit-based training on the level of general physical fitness of judokas, allowing us to adjust the last four months of the training process, if necessary. The study was conducted in accordance with the Declaration of Helsinki and approved by the Ethics Committee of the Armenian State Institute of Physical Culture and Sport (protocol code 2/2022).

Using personal and university connections of the first author, access was granted to recruit participants at a sports school. Prior to the study, the participants, their parents, and the sports school administration attended an informational session and provided oral consent to apply a new training approach during training and to participate in a social survey. The sports and cultural center regularly conducts medical evaluations, thus, prior to the study, all participants underwent a medical examination to ensure they were healthy and able to complete CrossFit-based training. 

### 2.2. Participants

Twenty male participants were randomly assigned by drawing lots to one of two groups: CrossFit-based training (experimental) and traditional training (control). There were ten athletes in each group with an average of three years of sports experience. Participants’ average age in both groups was 11 years old ± 0.64. Participants’ judo experience ranged from one to three years, so it was critical to adjust the content and moderation of the intervention to their level of training. This was done by adjusting the difficulty and/or load of each exercise for each participant. Mid-point data were available for 8 participants in the experimental group (i.e., one student switched from judo to music and one student was unavailable for mid-point testing) and 9 participants in the control group (i.e., one student took a temporary leave of absence).

### 2.3. Measures

Before introducing the training program, baseline testing was conducted by the researcher in the experimental and control groups with standard conditions and at least three attempts for each exercise. Immediately prior to testing, a standardized set of warm-up exercise was performed (e.g., neck, arm, and elbow circles, hip rotation, body circles, lunge tap, sit static lunge, low lunge, butterflies, jumping jacks, etc.). The following assessments were used to test motor skills in general physical fitness domains specifically for the ages of participants included in the study: maximum repetitions of pull-ups and leg raises on the Sweden wall (both for testing muscular strength), maximum repetitions of push-ups in 10 s (for testing speed-strength skill), long jump for maximum distance (test for lower body power), maximum repetitions of squats in 60 s and maximum repetitions of burpees in 30 s (both for testing speed-strength endurance), and time to complete 4 × 10 m shuttle run and 5 forward rolls (for testing coordination ability) [4,25,26]. Participants took one attempt at each exercise other than the long jump, to allow for sufficient recovery between tests. O Soto Gari and O Goshi throws were used to test Judo-specific physical fitness on a separate day. The first exercise (O Soto Gari) was suggested by Khomichev and Tarakanov [2], and the second exercise was chosen by study investigators, taking into account the participants’ sports experience.

In December 2021, a sociological survey was conducted among athletes of the experimental group to explore the athletes’ attitudes towards CrossFit-based training protocol using researcher-developed questions. The survey aimed to explore each athlete’s attitudes about the availability, uniqueness, and effectiveness of the CrossFit-based training. All athletes in the experimental group completed the sociological survey via Google Forms. There were nine questions in the survey (see Appendix A Appendix A). Questions were aimed to elucidate the athlete’s attitudes regarding the methods, means, and modalities which were used in the workouts (e.g., “Do you have difficulty doing CrossFit exercises?”, “What kind of exercises do you enjoy most in CrossFit-training?”, etc.). The survey was conducted in Armenian. Closed questions were used.

In order to control the athletes’ feeling towards the training load, their pulse rate was measured manually at the wrist during all the training sessions: before training, during training, and immediately after training. Athletes pressed the index and middle fingers of one hand on the opposite wrist counting the number of beats in 15 s and multiplied that number by four [27,28].

### 2.4. Training Programs

Participants in both groups had three days per week of judo training for 90 min. This was followed by 15–20 min of conditioning training. During the implementation of the training program, there were athletes who were absent from some workouts. In such cases, additional make-up trainings were conducted in accordance with the content of the training program.

During conditioning, the control group completed traditional methods of physical fitness training for judokas twice per week for 15–20 min. For example, they completed 100 repetitions of push-ups, sit-ups, and squats, or might complete 50 repetitions of pull-ups, 100 bridge exercises for the neck, and complete 10 rope climbs, etc.

CrossFit-based conditioning was implemented twice per week for 15–20 min in the experimental group after usual training. The CrossFit-based training was held at the end of each Judo training session, mainly after technical-tactical training. All CrossFit-based training was conducted by a trained sports professional with specific judo training.

CrossFit-based workouts included Tabata, “as many rounds/repetitions as possible” (AMRAP), and for-time workouts with a 21-15-9 repetition scheme performed at 60–70% intensity. Tabata training is a form of high-intensity interval training (HIIT) named after the Japanese scientist Dr. Izumi Tabata and lasts for only four minutes [29]. Each Tabata training interval involved 20 s of extremely intense exercise, followed by 10 s of rest. These exercises were primarily used to develop speed-strength endurance. A wide variety of push-ups, pull-ups, sit-ups, squats, metabolic exercises, handstand push-ups, shuttle runs, long jumps and other varieties of running, as well as exercises from weightlifting, such as cleans and snatches, were used.

The content of one Tabata training session was as follows:Push-ups with Tabata method (8 rounds of 20 s of exercise, then 10 s of rest).Squats with Tabata method.Sit-ups with Tabata method.

After each exercise, the athletes rested for one minute.

AMRAP workouts involved doing as many rounds and repetitions of a set of exercises as possible during a set amount of time, without designated rest. AMRAP workouts are often done as a circuit, which involves cycling through multiple types of exercises that use different parts of the body. AMRAP workouts can be performed with bodyweight exercises, weights, or cardiovascular exercises [30]. This method was used to develop both general endurance and strength endurance. 

The content of one AMRAP training session was as follows:

Do the maximum number of repetitions with the following exercises for 15 min:1 Leg raises on Sweden wall.Handstand push-ups (legs on the wall).Elevated heels.

Timed workouts included completing “21-15-9” repetitions of specific exercises. Athletes performed each movement in the workout at a fixed intensity (e.g., by using body weight) for 21 repetitions, then 15 repetitions, and then 9 repetitions for the fastest time [31]. These workouts were used to develop young judokas’ speed and speed-strength ability using exercises such as jumping and rolling. 

The content of one 21-15-9 training session was as follows:Burpees (21 repetitions, then 15 repetitions, and then 9 repetitions).Deep squats (21 repetitions, then 15 repetitions, and then 9 repetitions).Sit-ups (21 repetitions, then 15 repetitions, and then 9 repetitions).

After each circle, they completed 20 jumping jacks. 

### 2.5. Analysis

Data analyses were conducted with SPSS 27 (IBM, Corp., Armonk, NY, USA). After checking the data for assumptions (e.g., normality), independent samples *t*-test were used to examine for baseline differences between groups. Next, paired samples *t*-tests were conducted to examine changes from baseline to midpoint within each group. Cohen’s d was used for effect size, with 0.2 = small, 0.5 = medium, and 0.8 = large [32]. To compare changes in performance between groups, a group by time repeated measures analysis of variance (ANOVA) was conducted. Data are reported as mean ± standard deviation and statistical significance was set at *p* ≤ 0.05.

## 3. Results

No statistically significant differences were found between groups for any fitness or judo-specific fitness tests at baseline (all *p* > 0.05).

### 3.1. Within Group Comparisons

As shown in Table 1, the intervention group significantly improved from baseline to mid-point on push-ups repetitions (*t* = 2.72, *p* = 0.030), leg raise repetitions (*t* = 3.78, *p* = 0.007), and shuttle run time (*t* = 8.32, *p* < 0.001).

Table 2 shows changes in physical fitness for the control group participants. The only statistically significant difference was an improvement in the shuttle run time (*t* = 9.78, *p* < 0.001).

To assess changes in judo-specific physical fitness, athletes in both groups did “O Soto Gari” and “O Goshi” throws for 10 repetitions each. Table 3 shows changes in specific physical fitness for the experimental and control group participants. Athletes in the experimental group significantly decreased the time to complete both the O Soto Gari (*t* = 5.77, *p* = 0.001, 95% CI [−3.28, −0.77], Cohen’s d = 2.04) and the O Goshi throws (*t* = 2.5, *p* = 0.04, 95% CI [−1.70, −0.04], Cohen’s d = 0.89).

### 3.2. Between Group Comparisons

Between group comparisons using repeated measures ANOVA found a significant group by time interaction for push-ups, ƒ(1, 15) = 11.82, *p* = 0.004, due to the experimental group increasing their push-up repetitions and the control group decreasing their push-up repetitions from baseline to mid-point. A significant main effect was found for both groups for improving their shuttle run times, ƒ(1, 15) = 161.0, *p* < 0.001, although no significant differences between groups were found. A significant main effect was found for improvements in the leg raises, ƒ(1, 15) = 12.15, *p* = 0.003, as well as a significant group by time interaction, ƒ(1, 15) = 6.43, *p* = 0.023, where the intervention group improved significantly more than the control group. A significant main effect was found for improvements in the O Soto Gari, ƒ(1, 15) = 14.84, *p* = 0.002, with both groups decreasing the time needed to complete the throws. Finally, there was a significant group by time interaction for changes in the O Goshi, ƒ(1, 15) = 8.07, *p* = 0.012, where the experimental group decreased their time, and the control group increased their time needed to complete the throws.

On the sociological survey, all participants (*n* = 8, 100%) indicated participation in the CrossFit-based training with pleasure. Most preferred Tabata (85.7%) and for time “21-15-9” (42.9%) methods during CrossFit-based training. It was critical to understand if the participants felt bad or experienced pain during the CrossFit-based training. All participants mentioned that they rarely felt bad during training, but occasionally participants mentioned experiencing headaches (28.6%), dizziness (14.3%), and nausea (42.9%). These conditions were mostly associated with 21-15-9 (42.9%) and AMRAP (28.6%) training.

## 4. Discussion

This pilot study aimed to explore the effectiveness of a CrossFit training program to enhance the general and sport-specific physical fitness of 10–12-year-old judokas. The primary findings supported our hypothesis that CrossFit-based training would improve both general and sport-specific fitness more than traditional training methods for young judokas. Specifically, the experimental group significantly improved muscular endurance (i.e., push-ups and leg raise repetitions), speed (i.e., shuttle run), and judo-specific performance (i.e., O Soto Gari and O Goshi). The control group only significantly improved their speed (i.e., shuttle run). Between-group comparisons confirmed that the experimental group improved significantly more than the control group on muscular endurance (i.e., push-ups and leg raise repetitions) and judo-specific performance (i.e., O Goshi).

To compare results to previous research with 10-year-old judo athletes [4], we can note that the duration of CrossFit-based trainings was reduced for 10–12-year-old judokas. According to Khomichev’s [4] research, in weekly micro-cycle, CrossFit training lasted a total of 60 min, while with our training program, the maximum duration of these trainings was 40 min per week. We also minimized the number of Olympic weightlifting exercises to avoid unwanted side effects. One of the most important indicators throughout the implementation of the training program was the control of the athletes’ pulse rate. This was monitored by judo coaches to assess how athletes felt during the training by measuring the pulse rate so that they could determine the correct measurement of the training load.

While overall, participants reported liking the CrossFit-based training sessions, some reported experiencing pain and bad feelings during some sessions, mainly when using methods that required a high intensity. This information will help us change the training program’s content, e.g., reducing the number of 21–15-9 workouts, excluding some jumping exercises from the training program, and increasing time for rest between sessions. These results are similar to those with professional athletes who were more likely to have injuries and bad feelings in the presence of high-intensity training. For example, many authors note [33,34,35,36,37,38,39,40,41] that shoulder, knee, or spine injuries are common in CrossFit athletes, yet no injuries occurred among our participants. 

Previous research has shown that CrossFit-based training is effective physical training for improving combat sports younger-aged athletes’ motor skills [3,4,5,6]. The current study also demonstrated this effect. Specifically, we found that with a moderate load and accounting for age and experience, CrossFit-based training was effective for improving motor and sport-specific skills in 10–12-year-old judo athletes.

Study limitations include the specificity of the intervention population. As the pilot study aimed to reveal the uniqueness of using CrossFit to train 10–12-year-old male judokas, the obtained results can be related only to that age and sex and were conducted with a small sample size. As we mainly used exercises with body weight, the obtained results were predictable, as the most effective exercises, e.g., deadlift, shoulder press, and push press, were not used in the training system, considering the age of the athletes. We also used a simple manual measure of pulse rate, which could be subject to individual measurement error as compared to a device-based measurement, although the method has been considered a common and accessible method [27,28]. It is impossible to determine the underlying mechanisms that may have resulted in the fitness improvements observed in this study, which should be examined in future research [42]. As well, future research should examine changes in anthropometric characteristics along with changes in fitness.

As a result of the analysis of the mid-point test results, it was decided to make certain changes in the training program. Specifically, we will increase the number of exercises aimed at developing muscle strength and include running exercises in the content of training. We will increase the number of pull-ups exercises. With the improvement of weather conditions, outdoor workouts will be added, and finally, the number of running exercises will increase, which will be both aerobic and anaerobic in nature. Future research should identify the outcomes of CrossFit-based training in older and younger athletes as well as among athletes from other sports. Further research will be needed to use weightlifting, deadlift, squats, and more metabolic conditioning exercises in the process of physical training of older adolescents, and to find out the effect of the latter on changes in physical fitness of the athletes.

## Figures and Tables

**Table 1 jfmk-07-00083-t001:** Changes in physical fitness of 10–12-year-old judokas in the experimental group from baseline to mid-point test (*n* = 8).

Exercises	Baseline (M ± SD)	Midpoint (M ± SD)	95% CI	*d*
Pull-ups (repetitions)	4.6 ± 3.6	6.1 ± 4.8	−0.31, 1.15	0.43
Push-ups (repetitions in 10 s)	15.9 ± 4.0	20.0 ± 3.4 *	0.89, 1.79	0.96
Leg raises on Sweden wall (repetitions)	12.0 ± 1.03	19.0 ± 2.80 **	0.34, 2.03	1.34
Long jump (cm)	173.9 ± 14.0	178.1 ± 17.3	−0.34, 1.11	0.40
Shuttle run (s)	12.0 ± 0.5	10.5 ± 0.7 ***	−4.59, −1.27	2.94
Squats (repetitions in 60 s)	52.1 ± 3.1	53.8 ± 11.6	−0.55, 0.85	0.15
Burpees (repetitions in 30 s)	12.0 ± 1.6 ^a^	13.6 ± 1.1 ^a^	−0.01, 1.79	0.92

* *p* < 0.05, ** *p* < 0.01, *** *p* < 0.001; ^a^ = 7 participants.

**Table 2 jfmk-07-00083-t002:** Changes in physical fitness of 10–12-year-old judokas in the control group from baseline to mid-point test (*n* = 9).

Exercises	Baseline (M ± SD)	Midpoint (M ± SD)	95% CI	*d*
Pull-ups (repetitions)	5.7 ± 5.0	5.3 ± 5.4	−0.77, 0.54	0.12
Push-ups (repetitions in 10 s)	21.1 ± 3.2	18.1 ± 3.6	−1.43, 0.05	0.71
Leg raises on Sweden wall (repetitions)	11.9 ± 6.1	13.1 ± 7.2	−0.42, 0.91	0.25
Long jump (cm)	168.9 ± 14.8	174.0 ± 17.1	−0.11, 1.33	0.62
Shuttle run (s)	11.7 ± 0.5	10.4 ± 0.5 ***	−4.94, −1.55	3.26
Squats (repetitions in 60 s)	58.9 ± 5.7	58.3 ± 8.9	−0.73, 0.58	0.08
Burpees (repetitions in 30 s)	12.0 ± 1.6	13.6 ± 1.1	−1.24, 0.17	0.92

*** *p* < 0.001.

**Table 3 jfmk-07-00083-t003:** The level of judo-specific physical fitness at baseline and midpoint for 10–12-year-old judokas (*n* = 17).

Exercises	Experimental Group		Control Group	
	Baseline	Midpoint	Baseline	Midpoint
O Soto Gari	29.5 ± 3.9	22.7 ± 3.0 **	28.9 ± 3.8	27.0 ± 4.6
O Goshi	29.0 ± 4.8	25.0 ± 3.8 *	28.6 ± 5.0	28.9 ± 2.6

* *p* < 0.05, ** *p* < 0.01.

## Data Availability

Data are available from the authors upon request.

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
