# Peer review of "The Effectiveness of a CrossFit Training Program for Improving Physical Fitness of Young Judokas: A Pilot Study"

_jfmk, 2022, doi:10.3390/jfmk7040083_

Round 1

Reviewer 1 Report

This manuscript aimed to evaluate the effects of a CrossFit Training Program in the physical fitness of young judokas. Despite the importance of implementing innovative training methods as authors stated in the Introduction, manuscript lacks of key information to support this justification. Also, this manuscript has serious flaws that deeply reduce the quality of the study. Most of them are referred to methodological aspects. There is no sample size justification or calculation (or a flow diagram with possible dropouts and reasons). Also, eligibility criteria, recruitment process or blinding process, if it exists, have not been included. Some procedures lacks of scientific rigor. Key information about participants characteristics, testing procedures or even training program are not described enough. Consequently, this study is not replicable.

It is recommended that results and discussion are re-arranged to incorporate the aforementioned aspects. Results are not clear. Discussion is vague and lacks of interpretation and practical applications.

ABSTRACT

This section lacks of key information such as variables assessed during the testing session, sex of the participants, study design, etc.

What is the meaning of “positive effect”? positive for what? Authors need to state results in a more specific way, supported by statistical data.

INTRODUCTION

Authors state:  Such strategies are especially critical in a landscape 28 where competition rules may change, and athletes are highly competitive at increasingly 29 young ages.

What rules may change? Based on what? This sentence is confusing and does not add relevant information.

 Some experts believe that traditional training systems are outdated and em-30 phasize the requirement for obvious modernization using current methods and means

It is recommended to describe what traditional methods are referred and what type of modernization is required.

Authors are basing the justification of adding CrossFit as appropriate training method on students population, which have different training and physical fitness requirement. For students, maybe the performance is not the main objective. The same for police officers.

It is recommended to be specific according to the population that the present manuscript is centered on. Extending the fourth paragraph may be a good point, although it is recommended not to include details about specific studies in Introduction section.

Also, it would be interesting to include information about the importance of testing included and variables assessed in the study.

MATERIAL AND METHODS

Although authors state “Prior to the study the participants, their parents and the sports school administration 86 consented to apply a new training approach during training and to participate in a social 87 survey”, it is not clear how the procedure was. Please, inform if they were informed, how they were informed, and thus, if they signed an informed consent, as Declaration of Helsinki state. Also, the approval of Ethical Committee should be incorporated with the reference number and all the ethical requirements.

Participants:

How was the randomization process?

What was the eligibility criteria?

Authors state: so it was critical to adjust the content and moderation of the intervention to their level of training.

How intervention was adjusted for every participants? This sentence is really confusing. Please, clarify it.

Participants subheading lacks of important information such as sex of participants, anthropometric measures, hours of week training, level of competition, etc. It is recommended to include a table with these relevant characteristics (included years of experience in judo).

Other important information that should be included would be:

Sample size calculation?

Recruitment procedure?

Ways of contact with sample or school?

Blinding of evaluators, participants or those persons who implemented interventions?

The assessments that were used to test motor skills should be much more detailed. Previous warm-up? How was each test? How was measured? What measurements were collected? What tools were employed? What references validated these test and ensured that these testing tasks are appropriate? It is recommended to add a bullet proof for each exercise.

In reference to the sociological survey, based on what reference or study? How were these questions? What variables were extracted? How were it measured?

Again, measuring the heart rate in the wrist does not seem a rigorous method to be used. There are affordable methodologies to assess this variable in an easy way. It would be appropriate other type of fatigue monitoring such as Borg scale or other similar tools.

What means that control group completed “traditional methods”? please, describe in detail what training control group completed.

Was the “trained sports professional with specific judo training” external to investigation? Or was a member of research group?

It is recommended to include a detailed explanation of the CrossFit Training Program in a Table or Appendix.

Interpretation of effect size should be explained with more details and supported by a reference.

Additional variability statistical data are recommended to support results such as confidence interval 95%.

RESULTS AND DISCUSSION

Were there baseline differences between groups?? Normality of data?

It is recommended to include the between-group differences of the variables shown in tables 1 and 2 before including results from O Soto Gari and O Goshi testing and Table 3.

Table 3 have “*” sign, but it is needed to add the meaning. Statistical within-group or between-group differences?? Units of testing?

In general, tables should be understandable by itself. Please, check all abbreviations and explain it in the foot of the table.

Results regarding to the questionnaire should be detailed. What options they had? (to include in methods section)

Previous comments should be taken into account for the Discussion section (i.e. heart rate monitoring)

It is recommended to add discussion about main points of the manuscript in comparison with similar previous studies. Now, this section needs to be re-arranged and improved to incorporate interpretation according many factors such as age, performance, etc.

Line 226-228, it is information that belong to method section.

Line 229-234, maybe it would be more appropriate in the final part of the Discussion section.

Conclusion should be explicit, according to the purpose of the study.

Author Response

We thank the reviewer for their time spent reviewing and providing thoughtful comments to improve our manuscript. We have provided a point by point response in the attached file.

Reviewer 2 Report

Investigating the effects of training methods is a worthwhile line of research. The use of an experimental and control group over time is appropriate although you do need to consider the effects of placebo effects. The researchers find support for using cross-fit. However, as I read, I asked myself many questions and these questions need answering in the paper and thereby strengthen the rigour used in  the arguments for why doing the research was needed. Without strengthening the case for doing the research and justifying the methods, then the results are shallow and the article lacks credibility.

Key emerging questions:

Why only half a training cycle?

Why this age group in particular? 10-12 years?

Possible placebo effects – the new training raised motivation and it was this that led to improvement and so we can only cautiously say it was due to the specifics of cross-fit.

See Beedie, C., Benedetti, F., Barbiani, D., Camerone, E., Lindheimer, J., & Roelands, B. (2020). Incorporating methods and findings from neuroscience to better understand placebo and nocebo effects in sport. European Journal of Sport Science20(3), 313-325.

20 people is a small sample?

Why twice per week?

Why cross fit? The research you presented in support was from clearly older participants?

How was randomisation conducted and justify this approach?

Performance test was not specific to judo and so why take this approach and the performance test would seem to lend itself to cross-fit training?

Author Response

(The authors gave the same response as above.)

Reviewer 3 Report

This paper presents a study aimed to examine the effectiveness of a CrossFit-based training program to improve the physical fitness of 10- to 12-year-old judokas. The topic may have some interest but there are some clear weaknesses.

In particular, the most critical point is the small size of the two subsamples (consisting of 8 or 7 judokas each). Given this, I suggest turning your study into a pilot study.

The second critical point concerns applied statistics. With such small sample sizes, it is not recommended to use traditional statistical methods. Normality of the distribution would be essential in the case of small samples but, since it cannot be evaluated, I suggest using nonparametric statistics. Although the t-test can be applied to even small sample sizes, results may be unreliable if the assumptions of the t-test are not satisfied. The same is true for ANOVA.

Minor concerns:

-the sex of the participants was not specified

-line 120: change "heart rate" to "pulse rate".

-line 127: indicate in broad terms what the usual training consists of.

-Table 2: the symbol a is not shown in the note.

-line 223: change "heart rate" to "pulse rate".

-line 249 and subsequent lines: other limitations should be reported in the discussion: from the small subsample size to the assessment of pulse rate self-reported by participants.

-line 251: sex was not specified.

Author Response

(The authors gave the same response as above.)

Round 2

Reviewer 1 Report

I would like to congratulate authors for the great effort to improve the correctable aspects that manuscript had. All those points have been corrected, and manuscript has been considerably improved.

However, there are other flaws difficult to correct because it refer to methodological plan. As I stated, this investigation lacks of scientific rigor in many aspects. Many of these aspects cannot be justified only for being a pilot study or a pedagogical approach.

In consequence, there are many risk of bias due to important issues like: a very specific sample without randomization in the recruitment process, lack of control of key variables such as those related to anthropometric data, the lack of written consent, etc. In that sense, I am worried about the fact that the pedagogical nature of this study allowed participants not to be enroll if they did not really want.

This pilot study could be useful to plan an experimental study, taking into account during the design of the study those standard that would reduce risk of bias and would increase the quality of investigation, its internal and external validity. 

Author Response

I would like to congratulate authors for the great effort to improve the correctable aspects that manuscript had. All those points have been corrected, and manuscript has been considerably improved.

However, there are other flaws difficult to correct because it refer to methodological plan. As I stated, this investigation lacks of scientific rigor in many aspects. Many of these aspects cannot be justified only for being a pilot study or a pedagogical approach.

In consequence, there are many risk of bias due to important issues like: a very specific sample without randomization in the recruitment process, lack of control of key variables such as those related to anthropometric data, the lack of written consent, etc. In that sense, I am worried about the fact that the pedagogical nature of this study allowed participants not to be enroll if they did not really want.

This pilot study could be useful to plan an experimental study, taking into account during the design of the study those standard that would reduce risk of bias and would increase the quality of investigation, its internal and external validity. 

We thank the reviewer for their positive comments on our revisions. We agree that our study (like all studies) still has limitations. However, we have already mentioned that the division of the experimental and control groups was done by drawing lots. The children of the given age practicing in the sports school were divided into two groups based on the principle of lottery: experimental and control. Thus, there was random assignment to conditions.

Consent was obtained orally from both the parents and their children, thus they were able to decline participation then or at any time following.

Our research methods did not include the calculation of anthropometric data, as it was not related to the implementation of our goal of fitness training.

As for the children who stopped training, we should note that they simply moved to another field (music), that is, it was impossible to keep them in the research. All the other athletes participated in the entire experiment. In case of absence, additional individual training sessions were held.

We respectfully argue that it is important to publish pilot studies (e.g., https://www.mdpi.com/2411-5142/5/2/40 and https://www.mdpi.com/2411-5142/5/1/14 among others), along with fully powered ones to add to the research literature. Within those studies, it is important to acknowledge study strengths and weaknesses as we have in our discussion section.

Reviewer 2 Report

The authors have addressed  the reviewers comments. I would like to see arguments exposing the benefits of using cross-training dampened due to the relatively weak study design. The author have done a nice applied study that has encouraging results but it s possible to interpret results in different ways as the mechanisms through which changes occurred are not specified or tested,

Author Response

The authors have addressed  the reviewers comments. I would like to see arguments exposing the benefits of using cross-training dampened due to the relatively weak study design. The author have done a nice applied study that has encouraging results but it is possible to interpret results in different ways as the mechanisms through which changes occurred are not specified or tested.

We appreciate the reviewer’s comments and have included several study limitations in the discussion section. As a pilot study we were most interested in documenting changes in fitness rather than trying to identify the mechanisms underlying those changes and we have already included that as a limitation in the discussion.

Reviewer 3 Report

The authors modified the manuscript satisfactorily. I suggest only a rereading to avoid minor writing errors (see, for example, "The pilot study research" at line 124).

Author Response

The authors modified the manuscript satisfactorily. I suggest only a rereading to avoid minor writing errors (see, for example, "The pilot study research" at line 124).

We appreciate the attention to detail by the reviewer and have modified the wording in the manuscript to avoid minor writing errors, including that noted by the reviewer.

Round 3

Reviewer 1 Report

I thank authors for the effort to check those minor and modifiable aspects.